# Disease Progression and Serological Assay Performance in Heritage Breed Pigs following *Brucella suis* Experimental Challenge as a Model for Naturally Infected Feral Swine

**DOI:** 10.3390/pathogens12050638

**Published:** 2023-04-24

**Authors:** Vienna R. Brown, Ryan S. Miller, Courtney F. Bowden, Timothy J. Smyser, Nicholas A. Ledesma, Airn Hartwig, Paul Gordy, Aaron M. Anderson, Stephanie M. Porter, Kate Alexander, Zane Gouker, Thomas Gidlewski, Richard A. Bowen, Angela M. Bosco-Lauth

**Affiliations:** 1National Feral Swine Damage Management Program, USDA APHIS Wildlife Services, Fort Collins, CO 80521, USA; 2Centers for Epidemiology and Animal Health, USDA APHIS Veterinary Services, Fort Collins, CO 80521, USA; 3National Wildlife Research Center, USDA APHIS Wildlife Services, Fort Collins, CO 80521, USA; 4National Veterinary Services Laboratories, USDA APHIS Veterinary Services, Ames, IA 50010, USA; 5Department of Biomedical Sciences, Colorado State University, Fort Collins, CO 80521, USA; 6National Wildlife Disease Program, USDA APHIS Wildlife Services, Fort Collins, CO 80521, USA

**Keywords:** *Brucella suis*, culture, diagnostic assay, disease progression, experimental infection, feral swine, serology, swine brucellosis

## Abstract

Invasive feral swine (*Sus scrofa*) are one of the most important wildlife species for disease surveillance in the United States, serving as a reservoir for various diseases of concern for the health of humans and domestic animals. *Brucella suis*, the causative agent of swine brucellosis, is one such pathogen carried and transmitted by feral swine. Serology assays are the preferred field diagnostic for *B. suis* infection, as whole blood can be readily collected and antibodies are highly stable. However, serological assays frequently have lower sensitivity and specificity, and few studies have validated serological assays for *B. suis* in feral swine. We conducted an experimental infection of Ossabaw Island Hogs (a breed re-domesticated from feral animals) as a disease-free proxy for feral swine to (1) improve understanding of bacterial dissemination and antibody response following *B. suis* infection and (2) evaluate potential changes in the performance of serological diagnostic assays over the course of infection. Animals were inoculated with *B. suis* and serially euthanized across a 16-week period, with samples collected at the time of euthanasia. The 8% card agglutination test performed best, whereas the fluorescence polarization assay demonstrated no capacity to differentiate true positive from true negative animals. From a disease surveillance perspective, using the 8% card agglutination test in parallel with either the buffered acidified plate antigen test or the *Brucella abortus*/*suis* complement fixation test provided the best performance with the highest probability of a positive assay result. Application of these combinations of diagnostic assays for *B. suis* surveillance among feral swine would improve understanding of spillover risks at the national level.

## 1. Introduction

Disease surveillance systems are an important element of the animal health infrastructure used to detect and respond to disease events. Wildlife play a significant role in the dynamics of many economically important diseases, and are an increasingly crucial component of associated surveillance efforts [1,2,3]. Despite the importance of wildlife, challenges persist in implementing effective surveillance programs in wild populations [4,5]. One of the most important limitations is diagnostic tools, which are often unvalidated in wildlife [5]. Diagnostic tools designed to screen for infectious diseases in domestic animals frequently do not have the same levels of sensitivity and specificity when used in wild species [5]. This can complicate interpretation of surveillance results, reducing the sensitivity of the surveillance system.

One of the most important wildlife species for disease surveillance in the United States (U.S.) are invasive feral swine (*Sus scrofa*), commonly referred to as wild pigs or feral hogs [6]. Feral swine pose significant disease risks to animal agriculture and humans, with a large number (87%) of World Organisation for Animal Health (WOAH) swine pathogens potentially causing disease in livestock [7]. Many of these pathogens (i.e., *Leptospira* spp., *Salmonella* spp., hepatitis E virus, and *Brucella suis*) are zoonotic as well [7,8,9,10].

Among the pathogens carried and transmitted by feral swine, *B. suis* is considered among the most important due to its impact on domestic animals and risk to humans. Globally, feral swine serve as a significant reservoir for *B. suis*, which is a pathogenic intracellular Gram-negative bacteria and one of several species within the genus *Brucella* [11]. Both domestic and wild lineages of *S. scrofa* serve as the primary host for *B. suis*; however, infection has been documented in broad range of mammals [12,13,14,15,16,17]. In the U.S., *B. suis* biovars 1 and 3 are the predominant causative agents of swine brucellosis, whereas in Europe biovar 2 is commonly isolated from wild boar [11,18,19]. *B. suis* infection in swine is often characterized by osteoarticular lesions, lameness, mummified or weak piglets, orchitis, epididymitis, and abortion. Compared to other species of *Brucella*, *B. suis* presents unique disease management challenges, including prolonged intermittent bacteremia and shedding as well as venereal transmission [11].

In the U.S., swine brucellosis was eradicated from domestic pig production herds by 2011 [9 C.F.R Sect. 78.43] [20]; however, the pathogen is enzootic among feral swine [7,8,9,10]. Ongoing surveillance efforts conducted by the USDA-APHIS National Feral Swine Damage Management Program (NFSP) have demonstrated apparent seroprevalence rates ranging between 5.5% and 7% among feral swine in the U.S. [6], although estimates of the true prevalence may be much higher in certain populations of U.S. feral swine when accounting for imperfect detection [21]. The maintenance of *B. suis* in feral swine poses a sustained risk for disease spillover into domestic livestock, with pasture-raised pigs at greatest risk [22,23]. In the U.S., 57% and 77% of farms and domestic livestock, respectively, are co-located in areas with established feral swine populations [7]. Periodic spillover of *B. suis* from feral swine into domestic herds remains a concern, necessitating continued surveillance in domestic production systems [22,23]. Additionally, the NFSP opportunistically samples more than 6000 feral swine annually for swine brucellosis. These surveillance efforts rely on serological diagnostics to estimate the apparent seroprevalence of local populations, information which is then used to inform risk analyses related to spillover events in small-enterprise domestic swine herds and other livestock (USDA APHIS, unpublished data).

Diagnostic tools used to identify *B. suis* infection include bacteriological culture and serological assays; however, bacterial isolation from necropsy tissues or aborted material is considered the gold standard [24,25]. While bacteriological culture is feasible for diagnosing swine brucellosis in domestic production herds, collecting appropriate sterile tissue samples from large numbers of feral swine, as required for national scale surveillance, is impractical. Because of this limitation, whole blood samples that can be readily collected for serological assays are the preferred approach for conducting surveillance. However, serological assays can frequently have lower sensitivity and specificity, and few studies have validated serological assays for *B. suis* in feral swine. Currently serological assays for *B. suis* are considered unreliable at the individual animal level, though they are useful for determining relative rates of infection at the herd or population level [11,26]. Changes in diagnostic assay performance often occur during the course of infection due to variability in the kinetics, magnitude, or functionality of antibody response [27]. Importantly, the course of infection and changes in antibody response can be different between feral swine and domestic pigs due to differences in exposures to other pathogens or differences resulting from artificial selection of traits important for animal production [28].

The objective of the present study was to two-fold: first, to improve understanding of bacterial dissemination and antibody response of feral swine following infection with *Brucella suis*, and second, to evaluate potential changes in the performance of serological diagnostic assays throughout the course of infection. To address this objective, feral swine proxies were experimentally challenged with *B. suis* and serially euthanized over a 16-week period, during which time samples were collected to evaluate bacterial dissemination and antibody response measured using four diagnostic assays. We evaluated serological assay performance over time post-inoculation and determined the optimal combination of assays to use in serial or parallel assay interpretation of *B. suis*. Knowledge derived from this study is foundational for interpreting serological diagnostic results, from both domestic swine herds and naturally infected feral swine, and is useful when designing surveillance systems for feral swine.

## 2. Materials and Methods

### 2.1. Animals

For this experimental infection study, Ossabaw Island Hogs were selected as a domestic proxy for feral swine. The Ossabaw Island Hog breed of pigs was established by founders collected from the feral population inhabiting Ossabaw Island, Georgia, between 1986 and 1988. Thus, as a re-domesticated line of feral swine, Ossabaw Island Hogs represent the best available proxy for understanding disease progression in feral swine [29], as individuals from Ossabaw Island production herds can be certified to be free from disease, whereas disease-free status cannot be confirmed for wild-caught feral swine. Twenty-four mixed-sex (12 females, 11 castrated males, 1 intact male) and mixed-age (2 juveniles, 3 sub-adults, 15 adults, 3 unknown) pigs were obtained from a brucellosis-free swine herd maintained by a private producer in North Carolina. The pigs were transferred to the Animal Disease Laboratory at Colorado State University, a large animal BSL-3 Select Agent approved facility, where they were housed in 3.7 × 5.5 m challenge rooms and maintained on a commercial swine pellet ration with water provided ad libitum. None of the females were pregnant when inoculated. Pigs were split into two even groups of 12 based on strain of inoculation (strains described below), with males and females co-housed. Prior to inoculation, one castrated male was injured and deemed unfit for the study, providing final group numbers of 11 and 12 for strains 294 and 1330, respectively. Blood for serology was collected from each pig prior to inoculation as an additional confirmation that these animals had not been previously exposed to *Brucella suis.* Swine brucellosis serological diagnostics for both pre-inoculation and post-inoculation samples were performed by the National Veterinary Services Laboratories (NVSL) using the buffered acidified plate antigen test (BAPA), the 8% card agglutination test (card test), the fluorescence polarization assay (FPA) in tube format, and the complement fixation (CF) test [30,31,32,33].

### 2.2. Bacteria

Two biovar 1 strains of *Brucella suis* were used for this challenge to determine whether there was a detectable difference between a laboratory-propagated strain (1330 [NCTC 10316], NR-302, American Type Culture Collection, Biodefense and Emerging Infections Research Resources Repository [BIE Resources], NIAID, NIH, Manassas, VA) as compared to a wild-type field isolate (B17-0294, National Veterinary Services Laboratories, Ames, Iowa). Bacteria were grown to saturation in BHI broth supplemented with 5% horse serum and 0.1% glucose, harvested, and titrated on Farrell’s agar plates preceding inoculation.

### 2.3. Experimental Design

Prior to infection with *B. suis,* thermal microchips were implanted subcutaneously in the neck to collect data on body temperature (Destron-Fearing Bio-Thermo LifeChip). Pigs were challenged with subconjunctival dropwise instillation of 1 × 10^8^ *B. suis* suspended in PBS (50 µL per eye). Three pigs were euthanized (two 1330 strain, one 294 strain) and necropsied one week post-infection; four pigs (two per strain) were euthanized and necropsied at each of weeks two, four, eight, twelve, and sixteen. During the necropsies, the following samples were collected for culture: whole blood, lymph nodes (mandibular, retropharyngeal, parotid, mediastinal, mammary/inguinal, mesenteric), spleen, lung, liver, endometrium, vaginal swabs, and testicles. The aforementioned tissues, swabs, and whole blood were plated on Farrell’s plates immediately following necropsies and incubated for up to four days at 37 °C.

For animals euthanized at one, two, and four weeks post-infection (wpi), serological diagnostics were only conducted at one time point, specifically, from the blood collected at the terminal bleed during euthanasia. Starting at four wpi, serum was collected every four weeks for all surviving animals. Thus, for animals euthanized at eight wpi there were two time points with serological data (4 and 8 wpi), for animals euthanized at twelve wpi there were three time points with serological data (4, 8, and 12 wpi), and for animals euthanized at sixteen wpi there were four time points with serological data (4, 8, 12, and 16 wpi).

### 2.4. Clinical Observations

Temperatures were monitored daily for the first seven days post-inoculation. All animals were observed at least once daily for signs of clinical disease (fever, lethargy, anorexia, lameness) for the duration of the study.

### 2.5. Culture and Serology

Tissues were homogenized in PBS to a concentration of ~10% using a mixer mill and serial dilutions plated on Farrell’s agar plates. Colonies were counted three and four days later. DNA was extracted from at least one colony from each positive animal, and conventional PCR was used to confirm its identity as *Brucella* [34]. The sensitivity of the culture system was 100 cfu/g of tissue. Sera were filtered using 0.22 µm syringe filters (MilliporeSigma, Burlington, MA) affixed to syringes containing serum. Aliquots were cultured to confirm sterility, and the samples were shipped to NVSL for serologic testing.

All serum samples were submitted to NVSL for testing in parallel on the following brucellosis assays: BAPA, card test, FPA, and CF [30,31,32,33]. Serum samples for the BAPA and card test were incubated with their respective *Brucella abortus* antigen preparations and visually inspected for characteristic agglutination particles as compared with positive and negative serum controls [31,32]. Samples tested on FPA were mixed with FPA diluent, and a polarization measurement was captured to subtract background polarization [33]. Polarization was measured again after incubation with fluorescein-labeled antigen tracer. The average polarization of three negative controls was subtracted from the reading of each positive control and test sample to express the relative increase in polarization as delta millipolarization units (ΔmP). Samples were classified as follows: 0–10 ΔmP (negative); 11–20 ΔmP (suspect); and > 20 ΔmP (positive). All samples were tested in duplicate. The complement fixation test for swine was performed on each sample; the degree of hemolysis was observed visually and rated between 1 and 4 as an indicator of complement binding in the sample [30]. Reaction strengths above trace at 1:10 serum dilution were classified as positive, and the final titer was reported as the reaction strength at the well after the last dilution exhibiting complete complement fixation. The antigen used in all four serological diagnostic assays was derived from *Brucella abortus* strain 1119-3.

### 2.6. Statistical Analyses

#### 2.6.1. Differences in Tissues Infected

We evaluated potential differences between the *B. suis* strain, age, and sex of the animals for the types of tissues infected, total number of tissues infected, and level of bacterial tissue burden. Differences in the type of tissues infected were evaluated using Fisher’s exact probability test of binomial proportions. Potential differences in the total number of tissues infected and the level of bacterial tissue burden (bacterial colony counts) were evaluated using Fisher’s exact Poisson test. Statistical tests were implemented in the R statistical computing platform (version 4.1.1) [35].

#### 2.6.2. Diagnostic Assay Performance

We calculated four common metrics used to assess diagnostic assay performance to determine whether the assays performed differently for males versus females or for adults versus subadults. The metrics calculated included the sensitivity, specificity, predictive value (positive and negative), and diagnostic odds ratio. Sensitivity, that is, the true positive rate, is the probability of a diseased animal testing positive, and expresses the capacity of a diagnostic test to maximize true positives. Specificity, that is, the true negative rate, is the probability of a non-diseased animal testing negative, and encapsulates the capacity of a test to minimize false positives. Predictive value is a useful clinical metric for assessing diagnostic assay performance; positive predictive value is the probability of disease, that is, of returning a positive test, while negative predictive value is the probability of no disease, that is, of returning a negative test. The diagnostic odds ratio (DOR) describes the odds of a positive test in subjects with disease relative to the odds of a positive test in those without disease and provides a single measure that encapsulates the odds of an accurate test result. We used the Bayesian bootstrapping approach described by Rubin [36] to estimate the test performance metrics. Briefly, a posterior distribution was predicted for each metric for animals divided by sex and age and for all animals in the study by resampling the observed data 10,000 times. Bayesian bootstrapping was implemented using the bayesboot package and custom code in the R statistical computing platform (version 4.1.1) [35,37].

#### 2.6.3. Changes in Detection Probability through Time

To evaluate changes in the probability of detecting an infected animal post-inoculation, we used a Bayesian logistic model with repeated measures [38]. Such models relate the binary diagnostic assay result (positive versus negative) to the days post-inoculation, providing a prediction of the probability of a positive test result as a function of days post-inoculation. The individual animal was included as a grouping variable in the models to account for repeated measurements of certain animals. We included age, sex, and *B. suis* strain as predictors in the model to evaluate the potential effects on the time to detection. The models were used to predict the probability of a positive test result for each day post-inoculation from 1–113 days. A burn-in of 250,000 was used, and 20,000 samples were taken to predict the posterior distribution. Convergence of the models was evaluated using Gelman–Rubin and Geweke diagnostics [39,40]. Model fit was assessed using leave-one-out cross validation and Bayesian R^2^ [41,42]. Bayesian *p*-values were calculated for age, sex, and *B. suis* strain to determine any associations with the probability of a positive assay. Models were fitted and evaluated using the rstanarm package in the R statistical computing platform (version 4.1.1) [35,43].

#### 2.6.4. Optimal Combination of Diagnostic Assays

The optimal combination of diagnostic assays was evaluated for two common approaches used to determine whether an animal is positive. First, we considered interpreting diagnostic assays in series, where a subject is declared test-positive if both diagnostic assays are positive. In addition, we considered interpreting the diagnostic assays in parallel, where a positive outcome for either assay results in the subject being declared test-positive. Interpreting assay results in series increases diagnostic specificity and minimize false positives, while interpreting assay results in parallel increases diagnostic sensitivity, thereby maximizing true positives.

We considered all combinations of diagnostic assays and assigned animals as positive or negative using both serial and parallel assay interpretation. We evaluated each diagnostic assay combination using the Bayesian bootstrapping approach described above. Using this approach, and assuming independence among assays, we calculated the sensitivity, specificity, predictive value (positive and negative), and diagnostic odds ratio for each combination of assays. Additionally, we evaluated the performance of each combination of assays using receiver operating characteristic (ROC) curves to contrast the sensitivity and 1-specificity; these are commonly used to assess the overall diagnostic capacity of a diagnostic assay [44]. We used the area under the curve (AUC) produced by the ROC curve as a measure of overall assay accuracy [45]. We used the parametric approach from the ROCit package to calculate ROC curves and AUC values within the Bayesian bootstrapping approach [46].

## 3. Results

### 3.1. Differences in Tissues Infected and Serological Results

#### 3.1.1. Clinical Signs and Direct Culture

Throughout the study, none of the animals developed a fever or showed any clinical signs of disease. At necropsy, no gross lesions were observed in any of the pigs. Overall, bacterial culture revealed an increasing number of tissues infected and an increasing bacterial burden in those tissues until 4 wpi (Table 1, Figure 1). Beyond 4 wpi, the number of tissues infected and the bacterial burden in those tissues tended to decline; however, there was variation among animals beyond 4 wpi.

There were no statistically significant differences among males and females, adults and subadults, or the laboratory-propagated strain and wild-type field isolate in terms of the type and number of tissues infected (Appendix A, respectively). Vaginal swabs were frequently contaminated with other commensal bacteria; however, two female pigs (17%) had detectable *B. suis* from vaginal swab. *B. suis* was only detected in the endometrium of 25% of animals, and none after 4 wpi. *B. suis* was detected in the mammary and inguinal lymph nodes at all time points in 61% of animals. The testes of the single uncastrated male were culture-negative. Cranial lymph nodes (parotid, mandibular, retropharyngeal, mediastinal) were the most likely tissue to be infected with detectable *B. suis*, observed in 61-78% of animals, and up to 50% of animals at 16 wpi had detectable *B. suis* in these tissues. Caudal lymph nodes (mesenteric) were less likely to be infected, with only 39% of animals having detectable *B. suis* observed and none with detectable *B. suis* after 12 wpi. Organs (lung, liver, spleen) had detectable *B. suis* in 52–57% of animals and occurring at all time points.

There were differences in tissue bacterial burden across sex, age, and strain. Males had significantly higher bacterial burden when all tissues were considered together (Appendix A). The cranial lymph nodes (mandibular, retropharyngeal, and parotid) had higher bacterial burdens. Subadult animals had significantly higher bacterial burdens in the retropharyngeal and parotid lymph nodes (Appendix A). Adult animals had significantly higher bacterial burdens in the mediastinal and mesenteric lymph nodes. The strain of *B. suis* demonstrated differences in bacterial burden as well, with cranial lymph nodes tending to have higher burdens for strain 1330 (Appendix A). Strain 294 had higher bacterial burden in the mesenteric and mammary/inguinal lymph nodes as well as in spleen and lung tissues.

#### 3.1.2. Serology

Pre-challenge *B. suis* serological diagnostics were negative for all animals. All animals (100%) had positive serological results at 1 and 2 wpi on at least one assay. Animals with detectable antibodies against *B. suis* declined to 81.3%, 91.7%, and 75% at 4, 8, and 12 wpi, respectively. All animals (100%) had detectable antibodies at 16 wpi. These serologic results demonstrate substantial heterogeneity between samples collected from the same animal and at the same time point post-inoculation (Table 1, Figure 2). The capacity of serological assays to detect antibodies against *B. suis* declined from 1 to 16 wpi for all assays except the card test (Figure 2). Assay capacity to detect antibodies against *B. suis* declined at an average weekly rate of 7.2% (BAPA), 0.45% (card test), 9.0% (CF), and 10.2% (FPA). Two subjects had serological results that were anomalous. Subject 729 (euthanized at 12 wpi) was seropositive at 4 wpi using the card test but was seronegative according to all of the other assays and culture-negative at the time of euthanasia (Table 1, Figure 2). The second anomalous subject was 732, euthanized at sixteen weeks post-infection; this animal was seropositive on the card test at 8, 12, and 16 wpi but was culture negative at the time of euthanasia.

### 3.2. Diagnostic Assay Performance

#### 3.2.1. Single Assay Performance

The sensitivity of three assays (BAPA, card test, and CF) did not differ significantly across strata, and was predicted to be above 0.90 for all animals (Figure 3, Appendix A). The sensitivity of FPA was generally poor, with median values for all strata below 0.60. Specificity predictions had large variance and were not significantly different among the tests except for the card test, which had a median value above 0.90. However, the 95% credible interval for the card test overlapped with the three other assays. Positive predictive value had a generally large variation among strata, except for the card test, which had a median above 0.95 (Appendix A). All tests appeared to have the lowest positive predictive value for subadults, although the variance around the estimates was large. Negative predictive value for three of the assays (BAPA, card test, and CF) was generally above 0.90, with all assays having the highest negative predictive value for males. Negative predictive value for FPA was low for all strata and was below 0.60 for the stratum that included all animals (Appendix A). The diagnostic odds ratios (Figure 4, Appendix A) for adult animals tended to be higher compared to the other strata. Overall, the card test performed better than the other assays, and performed significantly better for adult female animals. All assays had low diagnostic odds ratios for subadult animals. FPA had the lowest performance for subadult animals, with an odds ratio close to 1, indicating that this assay provides no useful information and is equally likely to classify a subadult as positive regardless of the true condition of the animal.

#### 3.2.2. Changes in Detection Probability through Time

All logistic models demonstrated generally good predictive capacity (Appendix A). The probability of a positive test result after inoculation declined for all assays used in the study (Figure 5a). When using a single assay, only the card test maintained a detection probability above 0.95 after inoculation, and its detection probability remained above 0.90 until 101 days post-inoculation. Interpreting assays in parallel increased the detection probability (Figure 5b). Combinations that included the card test had probabilities of detection above 0.95 until 59 to 79 days post-inoculation. Interpreting assays in series reduced the probability of detection for all assay combinations. When interpreted in series, none of the assay combinations achieved a probability of detection above 0.90 after 8 days post-inoculation (Figure 5c).

The age and sex of the animal influenced the probability of detection post-inoculation differently (Appendix A). Relative to females, males were more likely to have a positive assay. Bayesian *p*-values for CF (0.031) were significant at the 0.05 level, while Bayesian *p*-values for FPA (0.0514) and BAPA (0.0633) were significant at the 0.1 level. Relative to adults, subadults were more likely to have a negative assay. However, only the card test (0.084) and CF (0.082) had significant Bayesian *p*-values at the 0.1 level. The strain of *B. suis* was not a significant predictor of probability of a positive assay at either the 0.05 or 0.01 level.

#### 3.2.3. Optimal Combination of Diagnostic Assays

The optimal combination of diagnostic assays differed when interpreting assay results in series or in parallel (Appendix A). When assays were interpreted in series or in parallel, the BAPA and CF assays together had the highest sensitivity and negative predictive value. The BAPA and CF assays together were the only combination that achieved a median sensitivity above 0.95 (Appendix A). Assay combinations that included the card test had the highest specificity values; however, none of the combinations achieved specificity values above 0.95. Similarly, assay combinations that included the card test had the highest positive predictive value. Overall, when assays were interpreted in parallel, the card test and BAPA assays together performed best, with a median AUC value of 0.987 (0.907–0.999; Appendix A). When assays were interpreted in parallel, three assay combinations (card test with BAPA, card test with CF, and BAPA with CF) had similar overall performance (Appendix A). The card test used in combination with CF, either in series or in parallel, achieved the highest median sensitivity (0.918 and 0.917, respectively) and the highest median specificity (0.936 and 0.937, respectively). Combinations of assays that included FPA had the lowest AUC values when assay results were interpreted in series or in parallel.

## 4. Discussion

Experimental *Brucella suis* infection of Ossabaw Island Hogs as a proxy for feral swine demonstrated variation in disease progression and serological assay performance. One diagnostic assay, the 8% card agglutination test, had the best overall performance, providing the highest sensitivity and specificity when used as a single assay. However, we observed variation in the performance of all assays, indicating that there may be limited value in using serological assays independently for feral swine. Using assays together in series or in parallel likely provides the best diagnostic utility. Evaluation of assay interpretation in series and in parallel indicated that several combinations of serological assays provided good predictive capacity for correctly identifying animals infected with *B. suis* and a high probability of accurately detecting *B. suis* post-infection. The interpretation of the card test and BAPA or the card test and CF in parallel resulted in good predictive performance.

Although evaluation of serological assay performance identified combinations of assays with good predictive capacity, diagnostic assay performance declined post-inoculation, with only the card test used with CF or BAPA achieving a probability of detection near or above 0.90 by the end of the study (16 weeks). Bacterial colony counts indicated variation in the tissues infected and the intensity of infection throughout the study. While *B. suis* strain was not a significant predictor of the probability of a positive assay, there were differences in bacterial tissue burden, which may influence the humoral immune response, potentially resulting in variation in serological assay performance. The percentage of animals with detectable antibodies declined post-inoculation for all assays except the card test. This may indicate an overall decline in the amount of antibodies present. The decline in the probability of detecting positive animals post-inoculation has implications for the interpretation of surveillance results. Animals that are newly infected with *B. suis* are more likely to have a true positive test result. This bias towards newly infected animals is most important for populations recently exposed to *B. suis*. Newly infected populations are not likely to have reached an equilibrium in transmission, meaning that there will be heterogeneity among infected and uninfected animals. Depending on the sampling design used, there may be significant bias in the resulting data. In populations that have had ongoing transmission of *B. suis*, the results of these serological assays are likely to be biased towards false negative results, producing prevalence estimates that are potentially low-biased if true and false detection probabilities are not accounted for in prevalence estimation [21,47]. Statistical methods are available to correct for errors in detection that result from diagnostic error [47]. The potential to misclassify animals that are not newly infected necessitates that an adequate number of animals be sampled from a given population to ensure that prevalence estimates can be corrected for detection error.

The potential to culture *B. suis* from tissue remains the most definitive test of infection, and as noted by the results of this study, most infected animals maintained a bacteriologic presence in some tissues until 16 wpi. All but two animals were culture-positive in one or more tissues at the time of necropsy. Cranial lymph nodes (parotid, mandibular, and retropharyngeal) were the most reliably positive tissues, and had the highest bacterial burdens, which is similar to the results of other studies [48]. Interestingly, the infection intensity in the mediastinal lymph node was different than in other cranial lymph nodes, approaching zero at 12 to 16 wpi. These findings are interesting in that a common natural route of infection is through the alimentary tract, occurring as swine ingest infected tissue commonly associated with abortions [49]. Because the animals used in this study were not inoculated via the alimentary tract, this may indicate that infection of these tissues is not a result of this route of exposure.

Differences among sex and age in both the bacterial burden in tissues and the probability of a serological positive test result in our study are different than in previous studies. Male animals had an overall higher bacterial tissue burden compared to females. Similarly, males were more likely to have a positive assay result compared to females. Animal age influenced bacterial burden as well, and potentially the probability of a positive assay result. Studies using surveillance data have found mixed associations for the effect of age and sex on seroprevalence [48,50]. However, interpreting risk factors associated with seroprevalence from surveillance data without accounting for detection error (i.e., diagnostic error) can result in erroneous conclusions or an inability to detect associations [47]. Considering that we found an association among sex, age, and the change in probability of a positive assay post-inoculation, there may be considerable bias in previously published associations involving age and sex.

Most animals seroconverted following infection and remained seropositive throughout the course of the study; however, there were two notable instances of anomalous results, both from female pigs infected with *B. suis* strain 1330. These findings mirror previously published results in naturally infected feral swine, in which *Brucella* was cultured from 77.5% of feral swine in an enzootically infected herd; however, only 54.1% of animals were seropositive [51]. It is important to note that the diagnostic sensitivities and, to a lesser degree, specificities reported in our study, are considerably higher than those reported by Pedersen et al. [48]. The animals in that study were naturally exposed and the temporal relationship between infection and euthanasia was unknown, which is likely to play an important role in test reliability. Aside from FPA, which fell below the previously published ranges, our estimates closely align with previously published diagnostic sensitivity and specificity ranges [48]. Taken together, these data suggest that the sensitivity of existing serological diagnostics for *Brucella* are insufficient for a single diagnostic assay to be used to determine disease status. For example, the poor diagnostic performance of the card test may explain the anomalous results for subjects 729 and 732, which were found to be seropositive by the card test at one (729) or multiple time points (732) throughout the study period but were culture-negative at euthanasia. Our results indicate that serological assays should be used in combination to determine disease status. Furthermore, serological data on *Brucella* using a single assay without correcting for detection error likely underrepresents the true infection status in wild populations [21].

Our study suggests that there are optimal combinations of diagnostic assays that would improve *B. suis* detection in feral swine and subsequent surveillance activities that can inform risk analyses in relation to spillover events. The 8% card agglutination test appeared to perform best, especially for adult animals. From a serosurveillance perspective, using the card test in parallel with either BAPA or CF is likely to provide the best performance and the highest probability of a positive assay result. The probability of detection declined for all assays, indicating that false negative results can be expected for animals that have not been recently infected. The FPA performed poorly as a single assay or in combination with other assays, and had no capacity to differentiate true positive and true negative animals, providing little or no information on historical exposure to *Brucella suis.*

While we identified potential combinations of assays, improved diagnostic assays remain needed for *B. suis*. The current international standards designate reagent strains of *B*. *abortus* for antigen production for use in brucellosis serodiagnostics, as the immunodominant antigens are believed to be highly conserved among *B*. *abortus*, *B*. *melitensis*, and *B*. *suis* [52]. Future work to improve the sensitivity of existing serological assays could explore the optimization of classification cutoffs, reconfiguration of the approved brucellosis surveillance testing sequence as applied to swine (both feral and domestic), and the evaluation of *B*. *suis* for antigen production.

## Figures and Tables

**Figure 1 pathogens-12-00638-f001:**
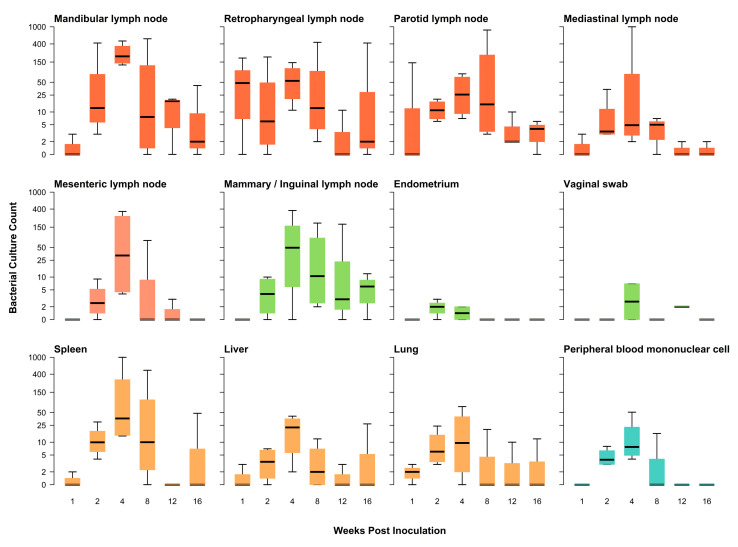
Bacterial culture counts for 12 tissues sampled from Ossabaw Island Hogs experimentally infected with *Brucella suis* and serially euthanized across a 16-week period. Cranial lymph nodes (dark orange) had among the highest tissue burdens, and these persisted until the end of the study. Organs (light orange) had tissue burdens extending the full length of the study as well. Cranial lymph nodes (dark orange); Caudal lymph nodes (peach); Female reproductive system (lime green); Organs (light orange); Whole blood (turquoise).

**Figure 2 pathogens-12-00638-f002:**
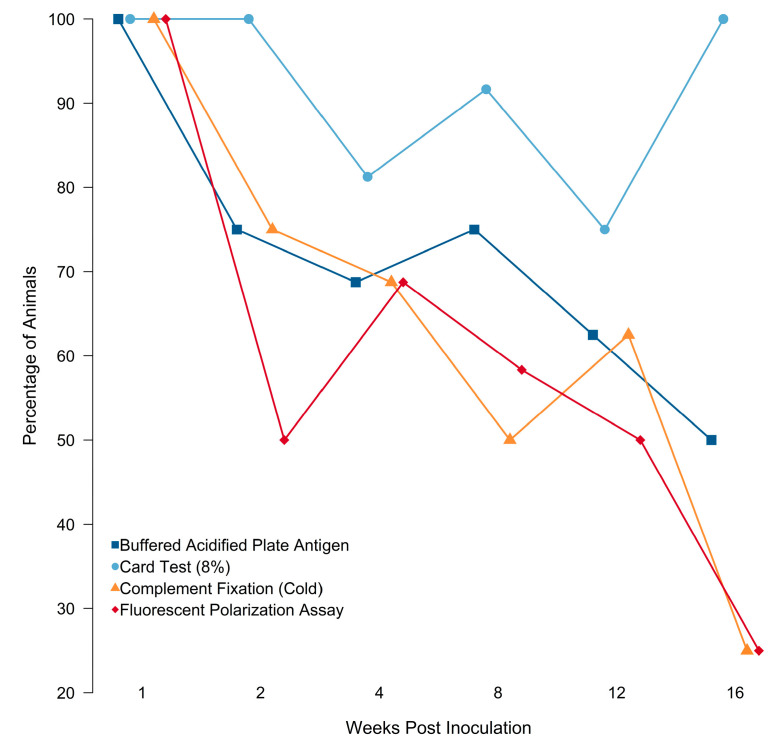
Percentage of Ossabaw Island Hogs with positive serological assay results for *Brucella suis* by week post-inoculation for the buffered acidified plate antigen test (BAPA), 8% card agglutination test (card test), fluorescence polarization assay (FPA) in tube format, and *Brucella abortus/suis* complement fixation (CF) test.

**Figure 3 pathogens-12-00638-f003:**
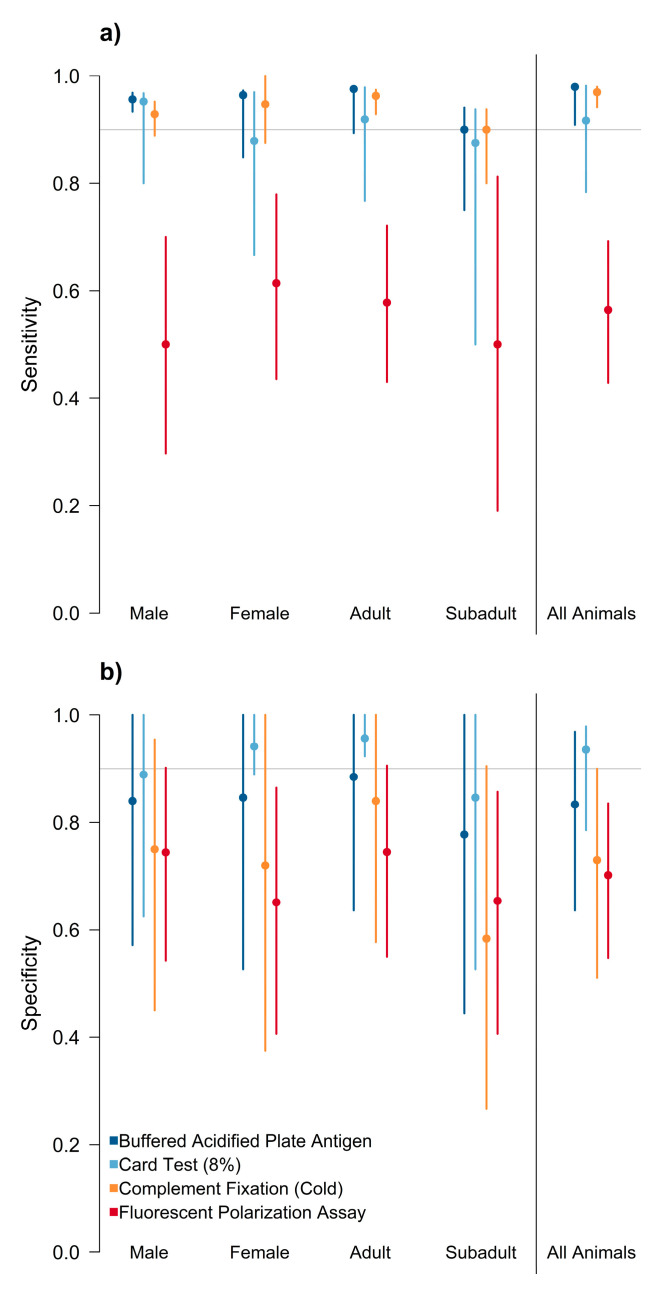
(**a**,**b**) Estimated sensitivity and specificity of the buffered acidified plate antigen test (BAPA), 8% card agglutination test (card test), fluorescence polarization assay (FPA) in tube format, and *Brucella abortus/suis* complement fixation (CF) test. Sensitivity was highest for the BAPA assay, while the card test had the highest specificity. All assays except for the FPA had point estimates of sensitivity above 0.90. FPA consistently had the poorest performance.

**Figure 4 pathogens-12-00638-f004:**
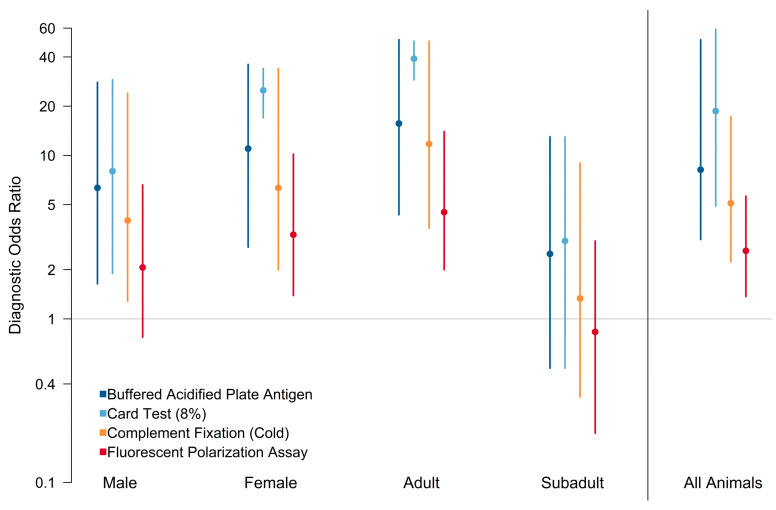
Diagnostic odds ratio for the buffered acidified plate antigen test (BAPA), 8% card agglutination test (card test), fluorescence polarization assay (FPA) in tube format, and *Brucella abortus/suis* complement fixation (CF) test. An odds ratio close to 1 indicates that the assay provides no useful information and is equally likely to classify an animal as positive regardless of the true condition of the animal. The diagnostic odds ratios for adult animals tended to be higher compared to the other strata. The card test performed better than other assays, and performed significantly better for adult female animals. All assays had low diagnostic odds ratios for subadult animals. FPA had the lowest performance for subadult animals.

**Figure 5 pathogens-12-00638-f005:**
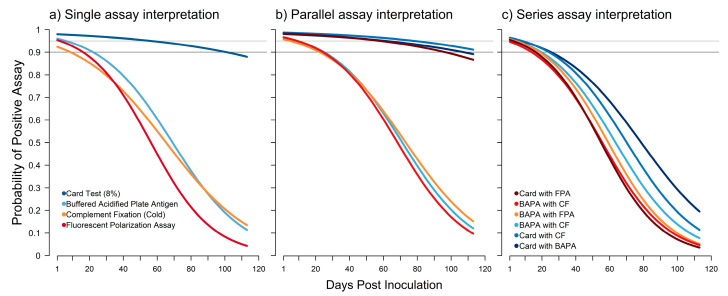
The change in probability of a positive assay result when using a single assay (**a**) or two assays interpreted either in parallel (**b**) or in series (**c**). Horizontal gray lines indicate 0.95 and 0.90 levels of probability. For single assays, the 8% card agglutination test (card test) was the only assay to consistently have a high probability of detecting antibodies. When interpreted in parallel, three combinations of assays generally performed well, with the card test used in combination with the *Brucella abortus/suis* complement fixation (CF) test having the highest probability across days post-inoculation.

**Table 1 pathogens-12-00638-t001:** Bacterial culture counts for Ossabaw Island Hogs experimentally infected with Brucella suis and serially euthanized across a 16-week period.

Pig #^e^	774	776	777	731	771	879	880	768	878	728	737	881	885	735	775	733	770	729	772	730	732	773	736	
**Wpi Euth ^b^**	1	1	1	2	2	2	2	4	4	4	4	8	8	8	8	12	12	12	12	16	16	16	16	
**Sex ^c^**	M	M	F	F	F	M	F	F	M	F	M	F	M	F	M	F	M	F	M	M	F	M	F	
**Age ^d^**	A	-	A	A	-	A	A	A	A	A	A	A	A	A	SA	A	A	A	SA	J	-	SA	J	
**Strain ^a^**	294	1330	1330	1330	1330	294	294	294	294	1330	1330	294	294	1330	1330	294	294	1330	1330	294	294	1330	1330	Percent Animals
**PBMC**	0	0	0	+	+	+	+	+	+	+	+	0	0	0	+	0	0	0	0	0	0	0	0	39
**Vaginal Swab**			0	0	0		0	+		0		0		0		+		0			0		0	17
**Endometrium**			0	+	+		0	0		+		0		0		0		0			0		0	25
**Testis**																				0				0
**Mandibular LN**	+ ^f^	0	0	+	++	+	+	++	++	++	++	0	+	+	+++	+	0	0	+	+	0	+	0	65
**Retropharyngeal LN**	++	0	+	+	++	0	+	++	+	+	+	+	+	+	++	+	0	0	0	+	0	++	0	70
**Parotid LN**	++	0	0	+	+	+	+	+	+	+	+	+	+	+	+++	+	+	0	+	0	0	+	+	78
**Mediastinal LN**	+	0	0	+	+	+	+	+++	+	+	+	0	+	+	+	0	0	0	+	0	0	+	0	61
**Mesenteric LN**	0	0	0	0	+	+	+	++	++	+	+	0	0	0	+	+	0	0	0	0	0	0	0	39
**Mammary/Inguinal LN**	0	0	0	+	+	+	0	++	+	0	+	+	+	+	++	++	0	0	+	0	0	+	+	61
**Spleen**	0	+	0	+	+	+	+	+++	+	+	+	+	0	+	++	0	0	0	0	0	0	+	0	57
**Liver**	0	+	0	+	0	+	+	+	+	+	+	0	0	+	+	0	0	0	+	0	0	+	0	52
**Lung**	0	+	+	+	+	+	+	+	+	0	+	0	0	0	+	0	0	0	+	0	0	+	0	52

^a^ *B. suis* inoculating strain: wild type = 294, laboratory propagated strain = 1330; ^b^ Weeks post-infection euthanized; ^c^ Sex: M = male, F = female; ^d^ Age class: A = adult, SA = subadult, J = juvenile, “-“ = unknown; ^e^ Pig #: Subject ID; ^f^ Bacterial burden in tissue: + [<100 colonies], ++ [100–500 colonies], +++ [>501 colonies].

## Data Availability

The data presented in this study are available within the article and Appendix A.

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
