# Peer review of "Disease Progression and Serological Assay Performance in Heritage Breed Pigs following *Brucella suis* Experimental Challenge as a Model for Naturally Infected Feral Swine"

_pathogens, 2023, doi:10.3390/pathogens12050638_

Round 1

Reviewer 1 Report

This is a very interesting paper since it concerns Brucella suis, a pathogen with zoonotic potential that affects wildlife and presents data gathered from an experimental challenge. Though, i have a few comments.

The “mixed age” should be explained in line 125, subsection 2.1.

“Tissues” should be mentioned in more detail in line 158, subsection 2.3

Sensitivity, specificity, predictive value and diagnostic odd ratios in lines 200-218, subsection 2.6.2, are explained too analytically. I believe that this kind of information doesn’t improve the quality of the paper.

In Table 1 (lines 270-271), there are three dashes (-) at the row of age. Please, explain what they stand for.

Please explain how the animal age influenced bacterial burden (line 455, section of discussion)

As mentioned in lines 313-318, subjects 729 and 732 yielded negative bacterial cultures, though seropositive results were reported at different times by using only card test. This finding could be explained as well by the fact that card test gave false positive results, and I believe that you should consider it in the discussion.  

I believe that "disease" in the title isn't correct, because clinical signs were not reported and thus, it could be replaced with a more appropriate word.  

Author Response

Dear Reviewer 1,

Thank you for your thorough and thoughtful review of our manuscript. Your suggestions have greatly improved our work and for that we are thankful. Please find below a point-by-point response to each of your suggestions.

Again, thank you!

Vienna Brown et al.

This is a very interesting paper since it concerns Brucella suis, a pathogen with zoonotic potential that affects wildlife and presents data gathered from an experimental challenge. Though, i have a few comments.

The “mixed age” should be explained in line 125, subsection 2.1.

Modified as suggested, see lines 125-126.

“Tissues” should be mentioned in more detail in line 158, subsection 2.3

We list the tissues in detail in the previous sentence. We’ve added “the aforementioned” to clarify, see line 159.

Sensitivity, specificity, predictive value and diagnostic odd ratios in lines 200-218, subsection 2.6.2, are explained too analytically. I believe that this kind of information doesn’t improve the quality of the paper.

We left this section unchanged as we believe it to be important to understand the methods used to evaluate the diagnostic assay performance.

In Table 1 (lines 270-271), there are three dashes (-) at the row of age. Please, explain what they stand for.

The dashes represent unknown age as data collection errors occurred during animal processing. We have added a description of the dashes to the table 1 footnote.

Please explain how the animal age influenced bacterial burden (line 455, section of discussion)

We have discussed this in the results section, please see lines 385-386.

As mentioned in lines 313-318, subjects 729 and 732 yielded negative bacterial cultures, though seropositive results were reported at different times by using only card test. This finding could be explained as well by the fact that card test gave false positive results, and I believe that you should consider it in the discussion.  

Modified as suggested, please see lines 479-482.

I believe that "disease" in the title isn't correct, because clinical signs were not reported and thus, it could be replaced with a more appropriate word.  

Please see lines 265-267 where we report the clinical signs associated with the infection – there were none. Accordingly, no changes were made to the title.

Reviewer 2 Report

Overall, this is a clear, concise, and well-written manuscript, that addresses an important topic due to the risk of disease transmission from feral swine to domestic pigs.  

The introduction is relevant and theory based. Sufficient information about the previous study findings is presented for readers to follow the present study. The content is technically sound, and overall, the methodology is well described.

Results section is clear and concise. However, supplementary files were not available and therefore not considered in this review.

The discussion answer to the original question, and suggests future research.

Minor comments:

-  Table 1 is not very easy to interpret. Maybe the authors can improve/simplify this table for better interpretation.

-  Lines 59, 61, 88, … – Replace “Brucella suis” by “B. suis” as already mentioned in the text.

-  Line 140 – The CFT test uses the B. abortus antigen, and is suitable for detection of antibodies to B. abortus, B. melitensis, and B. suis (as well as other smooth Brucella species). Please replace “Brucella abortus/suis complement fixation (CF) test” by “complement fixation (CF) test”.

-  Line 158 and 159 – the authors refer that “Tissues, swabs, and whole blood were plated on Farrell’s plates immediately following necropsies and incubated for up to four days at 37◦ C”. Why not incubate for, at least, seven-ten days? The incubation period for four days seems to me to short, and probably decreased the isolation’s sensitivity.

-  Line 419: correct “.90” by “0.90”

Author Response

Dear Reviewer 2,

Thank you for your thorough and thoughtful review of our manuscript. Your suggestions have greatly improved our work and for that we are thankful. Please find below a point-by-point response to each of your suggestions.

Again, thank you!

Vienna Brown et al.

Overall, this is a clear, concise, and well-written manuscript, that addresses an important topic due to the risk of disease transmission from feral swine to domestic pigs.  

The introduction is relevant and theory based. Sufficient information about the previous study findings is presented for readers to follow the present study. The content is technically sound, and overall, the methodology is well described.

Results section is clear and concise. However, supplementary files were not available and therefore not considered in this review.

The discussion answer to the original question, and suggests future research.

Minor comments:

-  Table 1 is not very easy to interpret. Maybe the authors can improve/simplify this table for better interpretation.

We fully agree with this comment. While the table is busy, we believe it is helpful to have the metadata reported with the bacterial culture data. In prior versions, the table was less scrambled as it was presented in landscape format as opposed to portrait. We will work with the editorial office to ensure that this table is legible.

-  Lines 59, 61, 88, … – Replace “Brucella suis” by “B. suis” as already mentioned in the text.

Modified as suggested.

-  Line 140 – The CFT test uses the B. abortus antigen, and is suitable for detection of antibodies to B. abortus, B. melitensis, and B. suis (as well as other smooth Brucella species). Please replace “Brucella abortus/suis complement fixation (CF) test” by “complement fixation (CF) test”.

Modified as suggested.

-  Line 158 and 159 – the authors refer that “Tissues, swabs, and whole blood were plated on Farrell’s plates immediately following necropsies and incubated for up to four days at 37◦ C”. Why not incubate for, at least, seven-ten days? The incubation period for four days seems to me to short, and probably decreased the isolation’s sensitivity.

Positive control cultures (B. suis) were plated along with the experimental samples, so any colonies that didn't grow in the same time frame as the positive control (2-3 days) would be unlikely to be Brucella suis. 

-  Line 419: correct “.90” by “0.90”

Modified as suggested.
